# Metal–Organic Frameworks (MOFs) Derived Materials Used in Zn–Air Battery

**DOI:** 10.3390/ma15175837

**Published:** 2022-08-24

**Authors:** Dongmei Song, Changgang Hu, Zijian Gao, Bo Yang, Qingxia Li, Xinxing Zhan, Xin Tong, Juan Tian

**Affiliations:** 1School of Chemistry and Material Science, Guizhou Normal University, Guiyang 550001, China; 2Key Laboratory for Functional Materials Chemistry of Guizhou Province, Guiyang 550001, China

**Keywords:** metal–organic framework, oxygen reduction reaction, oxygen evolution reaction, Zn–air battery

## Abstract

It is necessary to develop new energy technologies because of serious environmental problems. As one of the most promising electrochemical energy conversion and storage devices, the Zn–air battery has attracted extensive research in recent years due to the advantages of abundant resources, low price, high energy density, and high reduction potential. However, the oxygen reduction reaction (ORR) and oxygen evolution reaction (OER) of Zn–air battery during discharge and charge have complicated multi-electron transfer processes with slow reaction kinetics. It is important to develop efficient and stable oxygen electrocatalysts. At present, single-function catalysts such as Pt/C, RuO_2_, and IrO_2_ are regarded as the benchmark catalysts for ORR and OER, respectively. However, the large-scale application of Zn–air battery is limited by the few sources of the precious metal catalysts, as well as their high costs, and poor long-term stability. Therefore, designing bifunctional electrocatalysts with excellent activity and stability using resource-rich non-noble metals is the key to improving ORR/OER reaction kinetics and promoting the commercial application of the Zn–air battery. Metal–organic framework (MOF) is a kind of porous crystal material composed of metal ions/clusters connected by organic ligands, which has the characteristics of adjustable porosity, highly ordered pore structure, low crystal density, and large specific surface area. MOFs and their derivatives show remarkable performance in promoting oxygen reaction, and are a promising candidate material for oxygen electrocatalysts. Herein, this review summarizes the latest progress in advanced MOF-derived materials such as oxygen electrocatalysts in a Zn–air battery. Firstly, the composition and working principle of the Zn–air battery are introduced. Then, the related reaction mechanism of ORR/OER is briefly described. After that, the latest developments in ORR/OER electrocatalysts for Zn–air batteries are introduced in detail from two aspects: (i) non-precious metal catalysts (NPMC) derived from MOF materials, including single transition metals and bimetallic catalysts with Co, Fe, Mn, Cu, etc.; (ii) metal-free catalysts derived from MOF materials, including heteroatom-doped MOF materials and MOF/graphene oxide (GO) composite materials. At the end of the paper, we also put forward the challenges and prospects of designing bifunctional oxygen electrocatalysts with high activity and stability derived from MOF materials for Zn–air battery.

## 1. Introduction

In recent years, great progress has been made in economic development and science technology. New technologies and inventions have brought great convenience to modern society. However, these advanced technologies rely on the consumption of non-renewable fossil energy such as coal and oil. While fossil energy promotes social and economic development, it also causes serious environmental problems, such as the greenhouse effect, acid rain, and smog. The extreme depletion of fossil fuels and the harm to the environment have prompted researchers to develop more efficient, renewable, and clean energy conversion and storage equipment [1]. The metal–air battery is a new energy technology, which has the characteristics of rich raw materials, low cost, high safety, and environmental friendliness. It can be divided into the Li-O_2_ battery, Na-O_2_ battery, Zn–air battery, Al–air battery, and Mg–air battery (Figure 1) [2]. Among them, the rechargeable Zn–air battery is considered to be one of the most promising metal–air battery technologies because of its low cost, high energy density, environmentally friendly, safe operation, and renewable utilization [3,4].

Zn–air batteries usually contain four main parts: zinc anode, electrolyte, separator, and air cathode. Zn anodes generally use pure zinc metal as the active material, and zinc oxidation occurs during discharge. Zn metal has the advantages of low equivalence weight, reversibility, high specific energy density, easy accessibility, low toxicity, low cost, and no significant corrosion in water and alkaline media, making Zn–air battery a promising energy storage device [3,5]. The electrolyte plays a role in ion migration during the reaction, and its conductivity is a key factor affecting the ohmic resistance of the cell [6]. Zn–air batteries mostly work in alkaline media, especially KOH, because of their excellent ion conductivity (K^+^, 73.50 Ω^−1^ cm^2^/equiv), high oxygen diffusion coefficient, low viscosity, and excellent activity on both zinc anodes and air cathodes [7]. For highly active metal–air cells that are unstable in aqueous solutions, such as Li-O_2_ and Na-O_2_ batteries, non-aqueous aprotic electrolytes are usually required [8]. The main function of the separator is to keep apart the two different electrolytes and transport hydroxyl ions (OH^-^) from the air cathode to the zinc anode [2,8]. The separator must meet the basic requirements of inhibiting the growth of zinc dendrites, being stable in alkaline solutions, having low ionic resistance and high electrical resistance, as well as against the corrosive electrolyte and oxidation [2,4,9].

Air cathodes usually consist of an electrocatalytic layer and a gas diffusion layer, both of which serve to reduce the electrode overpotential and enhance the diffusion of oxygen between the ambient air and the catalyst surface, respectively [2]. The electrochemical reactions at the air cathode are the key to drive oxygen reduction reaction (ORR) during discharge and oxygen evolution reaction (OER) during charge [10,11,12,13]. However, the kinetics of the multi-electron transfer process of OER and ORR the sluggish O_2_ reduction kinetics. Currently, Pt-based catalytic particles exhibit the best catalytic performance for ORR [14] and either IrO_2_ or RuO_2_ for OER [15,16], but these rare and expensive noble metal nanomaterials can only selectively catalyze ORR or OER with poor long-term stability single function. Thus, the rational design of rich resources, high efficiency, and durability of non-noble metal bifunctional electrocatalysts is the key to improve the kinetics of ORR and OER reactions and accelerate the commercial application of Zn–air battery [17,18,19].

Metal–organic frameworks, also known as porous coordination polymers, are porous crystalline materials composed of organic ligands bridging inorganic metal ions/clusters. MOFs have a series of advantages such as adjustable porosity, highly ordered pore structure, low crystal density, and large specific surface area [20,21]. The organic ligand of MOFs can be converted into a heteroatom-doped carbon matrix with conductivity, and the metal species can directly interact with the heteroatom to improve the catalytic activity [22]. MOFs have been one of the hottest topics in materials chemistry and other disciplines and are widely used in gas storage/separation [23,24,25], multiphase catalysis [26,27,28], biomedicine [29,30,31], energy conversion [32,33,34], and so on, since they were first report in 1995 [35]. MOF-derived functional materials are the most ideal precursors for electrocatalysts, which are widely used in electrochemistry due to their unique structure and chemical composition.

In this review, the structure and working principle of Zn–air batteries are introduced firstly, as well as the basic principles and evaluation parameters of ORR and OER on air cathode. After that, some strategies to improve the activity and stability of MOF-derived ORR/OER electrocatalysts are summarized. The strategies were employed containing: (i) transition metal catalysts derived from single and multiple transition metal-doped MOF, (ii) metal-free catalysts derived from heteroatom-doped MOFs and MOFs/graphene oxide (GO) composites. In addition, the current challenges and prospects are discussed in the development of MOFs-derived electrocatalysts and Zn–air battery.

## 2. Catalytic Mechanism in Zinc-Air Battery

### 2.1. The Structure and Working Principle of the Zn–Air Battery

The basic structure of the Zn–air battery is shown in Figure 2. During discharge, oxygen is reduced to hydroxyl ions (OH^−^) at cathode side. Zinc react with OH^−^ migrate from cathode to form soluble zincate ions ${\mathrm{Zn}(\mathrm{OH})}_{4}^{2-}$, which can be decomposed to form the insoluble substance zinc oxide (ZnO). On the opposite, the OER on the air cathode involves the reverse process of the ORR during charge. The reaction process as follows: 

Discharge

Anode:(1)$${\mathrm{Zn}+4\mathrm{OH}}^{-}\;\to {\;\mathrm{Zn}(\mathrm{OH})}_{4}^{2-}{+2e}^{-}{\;(\varphi }^{0}=1.245\;V\;\mathrm{vs}.\;\mathrm{RHE})\;$$
(2)$${\mathrm{Zn}(\mathrm{OH})}_{4}^{2-\;}\to {\;\mathrm{ZnO}+H}_{2}{O+2\mathrm{OH}}^{-}$$

Cathode:(3)$$O_{2}{+2H}_{2}{O+4e}^{-}\;\to {\;4\mathrm{OH}}^{-}{\;(\varphi }^{0}=0.401\;V\;\mathrm{vs}.\;\mathrm{RHE})\;$$

Overall reaction:(4)$${2\mathrm{Zn}+O}_{2}\;\to \;2\mathrm{ZnO}\;(E=1.646.\;V\;\mathrm{vs}.\;\mathrm{RHE})$$

Charge

Anode:(5)$${\mathrm{ZnO}+H}_{2}{O+2\mathrm{OH}}^{-}\;\to {\;\mathrm{Zn}(\mathrm{OH})}_{4}^{2-}$$
(6)$${\mathrm{Zn}(\mathrm{OH})}_{4}^{2-\;}{+2e}^{-}\;\to {\;\mathrm{Zn}+4\mathrm{OH}}^{-}{\;(\varphi }^{0}=-1.245\;V\;\mathrm{vs}.\;\mathrm{RHE})$$

Cathode:(7)$${4\mathrm{OH}}^{-}\;\to {\;O}_{2\;}{+2H}_{2}{O+4e}^{-}{\;(\varphi }^{0}=-0.401\;V\;\mathrm{vs}.\;\mathrm{RHE})$$

Overall reaction:(8)$$2\mathrm{ZnO}\;\to {\;2\mathrm{Zn}+O}_{2}\;(E=-1.646\;V\;\mathrm{vs}.\;\mathrm{RHE})$$

According to the reaction process during discharge, the standard potential of Zn–air battery is 1.646 V ($E=\varphi_{\mathrm{Cathode}}^{0}-\varphi_{\mathrm{Anode}}^{0}$). However, the practical working voltage of the Zn–air battery is less than 1.646 V due to the internal loss of the battery by ohmic and concentration loss. During battery charging, a charging voltage of 2 V or even higher is required to satisfy the electrochemical reactions of the charging process. The overpotential of oxygen reaction on the air cathode and the formation of dendrites on the zinc anode are the reasons for significant deviation from the equilibrium value [36,37,38]. The slow kinetics of ORR and OER on the air cathode will cause activation loss. The poor flow of electrolytes and high overpotential will affect the cycle life of the battery [4,37]. In the following section, the reaction mechanism of ORR and OER are briefly introduced.

### 2.2. ORR Reaction Mechanism and Evaluation Parameters

The oxygen molecules are reduced during the discharge. The process of ORR mainly includes: diffusion and adsorption of O_2_ on the catalyst surface, electron transformation from the anode to the adsorbed O_2_, weaken and cleavage of the O=O bond, and migration of the generated OH^−^ in the electrolyte. ORR process is a multi-electron reaction in which complex oxygen-containing intermediates are formed. Depending on the type of adsorption, ORR may be a four-electron pathway or a two-electron pathway. Two-electron pathway the latter generates hydrogen peroxide ions, which could be further reduced or undergo disproportionation reactions (Equations (9)–(12)). Hydrogen peroxide generated by the two-electron process has high oxidizability, which not only reduces the efficiency of ORR, but also corrodes the catalyst materials [10]. Therefore, the direct four-electron pathway with produced H_2_O only avoiding the generation of peroxide species is preferred.

Alkaline: 

Four-electron pathway: (9)$$O_{2}{+2H}_{2}{O+4e}^{-}\;\to {\;4\mathrm{OH}}^{-}{\;(\varphi }^{0}=0.401\;V\;\mathrm{vs}.\;\mathrm{RHE})$$

Two-electron pathway:(10)$$O_{2}{+H}_{2}{O+2e}^{-}\to {\;\mathrm{HO}}_{2}^{-}{+\mathrm{OH}}^{-}$$
(11)$${2\mathrm{HO}}_{2}^{-}{+H}_{2}{O+2e}^{-}\to {\;3\mathrm{OH}}^{-}\;$$
(12)$${2\mathrm{HO}}_{2}^{-}\;\to {\;2\mathrm{OH}}^{-\;}{+O}_{2}\;$$

The onset potential (E_0_), half-wave potential (E_1/2_), Tafel slope and electron transfer number (n) are widely used to evaluate the catalyst ORR performance. These values are usually obtained by electrochemical techniques such as linear scanning voltammetry (LSV) and cyclic voltammetry (CV). LSV and CV are generally conducted in O_2_-saturated electrolytes with the catalyst dispersed on a rotating disk electrode (RDE) or rotating ring disk electrode (RRDE) [39]. The E_0_ reveals the electrode potentials from which ORR start, and E_1/2_ means that the reaction current density is equal to half of the limiting current density, respectively. These values depend on the inherent nature of the catalysts and can be used to evaluate their catalytic activities. The more positive E_0_ and E_1/2_ mean the smaller overpotential and the higher energy conversion. The Tafel slope reflects the kinetic rate of the ORR process, the smaller the Tafel slope is, the faster the ORR kinetics of the catalyst has [40]. The n mainly indicates whether the ORR process is dominated by two-electron or four-electron processes. It can be calculated by the Koutecky–Levich equation using LSV data obtained from RDE (Equation (13)) or by the yield of H_2_O_2_ measured by RRDE (Equations (14) and (15)).
(13)$$\begin{array}{c} \frac{1}{J}=\frac{1}{J_{L}}+\frac{1}{J_{K}}=\frac{1}{{B\omega }^{1/2}}+\frac{1}{J_{K}}\\ B=0.62{\mathrm{nFC}}_{0}{{(D}_{0})}^{2/3}\upsilon^{-1/6} \end{array}$$
where J, J_L_, and J_K_ are the measured current density, diffusion limiting current density, and kinetic current density, respectively, n is the number of electron transfer, F is the Faraday constant (F = 96,485 C/mol), C_0_ is the concentration of O_2_ in the electrolyte, D_0_ is the diffusion coefficient of O_2_ in the electrolyte, υ is the kinetic viscosity of the electrolyte, and ω is the rotation speed of the electrode.
(14)$${\%H}_{2}O_{2}=200\frac{I_{R}/N}{I_{D}{+I}_{R}/N}\;$$
(15)$$n=4\frac{I_{D}}{I_{D}{+I}_{R}/N}\;$$
where I_D_ and I_R_ are the disk current and ring current, respectively, and N is the collection efficiency of the ring.

### 2.3. OER Reaction Mechanism and Evaluation Parameters

As the reverse reaction of ORR, OER is also an important part of Zn–air battery. H_2_O is oxidized to O_2_ during the reaction, the complete reaction mechanism can be summarized with Equation (7), where the asterisk super-index (*) and (a) refer to active site and adsorbed on it, respectively.

Alkaline:$${\mathrm{OH}}^{-}+*\;\to {\;\mathrm{HO}}^{*}{(a)+e}^{-}$$
$${\mathrm{HO}}^{*}{(a)+\mathrm{OH}}^{-}\to {\;O}^{*}{+H}_{2}{O+e}^{-}$$
$$\;O^{*}{+O}^{*}\to {\;O}_{2}+2*\;(\mathrm{process}\;\mathrm{one})$$
$$O^{*}{+\mathrm{OH}}^{-}\to {\;\mathrm{HOO}}^{*}{+e}^{-}$$
$$\;{\mathrm{HOO}}^{*}{+\mathrm{OH}}^{-}\to {\;O}_{2}{+H}_{2}{O+*+e}^{-}\;(\mathrm{process}\;\mathrm{two})$$

In alkaline solution, OH^−^ adsorbs on the active site (*) to release electrons and generate HO*(a), and then HO*(a) react with OH^−^ to generate O*. O* generates O_2_ in two ways: (1) the direct coupling of two adjacent O* to form O_2_ molecules (process one); (2) the reaction of O* and OH^−^ to form the intermediate HOO*, which continues to react with OH^−^ to form O_2_ molecules (process two) [41]. The process one with O_2_ generated from HOO* is easier than process one due to its low thermodynamic potential barrier [42].

Overpotential (ƞ), Tafel slope (b), turnover frequency (TOF), and exchange current density (i_0_) are the main parameters to evaluate the electrocatalytic performance of catalysts for OER. When the electrode reaction is unbalanced, the potential difference between the electrode potential and the reaction equilibrium potential (1.23 V vs. RHE) is called the overpotential (ƞ). The lower the ƞ value is, the better the electrocatalytic activity is. Tafel slope (b) indicates the speed of overpotential is increasing with current density, as shown in Equation (16).
(16)ƞ=a+b logi
where a, b are both Tafel constants, i is the current density, and a lower value of b indicates better electrocatalytic performance. 

Using Tafel slope only to evaluate the catalytic performance is not accurate since there will be non-negligible errors in the value obtained from the graph. TOF is the conversion rate per unit of catalytic activity of the reactant molecule, which is used to measure catalyst activity, as shown in Equation (17).
(17)$$\mathrm{TOF}=\frac{{j\text{ × }N}_{A}}{\mathrm{AnF}\Gamma }$$
where j is the current, N_A_ is the Avogadro constant (N_A_ = 6.022 × 10^23^ mol^−1^), A is the geometric area of the electrode, n is the number of reaction electrons, F is the Faraday constant (F = 96,485 C/mol), and Γ is the surface concentration. The exchange current density reflects the inherent efficiency of electron transfer between the electrode and the electrolyte [43].

MOF-derived catalysts are one of the most promising bifunctional catalysts for ORR and OER. MOF-derived transition metal catalysts and MOF-derived non-metallic catalysts are illustrated in the following parts according to the structural characteristics of MOF materials. We would like to build connections between performance of catalysts and their preparation strategies, and give some suggestions that will be helpful for the future development of MOF-derived ORR/OER catalysts. As shown in Figure 3, we present a classification diagram of MOF-derived for air cathode catalysts.

## 3. MOF-Derived Non-Precious Metal Catalysts

In 1964, Jasinski [44] demonstrated for the first time that cobalt phthalocyanine can catalyze ORR in alkaline media. In 2009, Lefèvre et al. [45] prepared a Fe-N/C catalyst using black pearl 2000, 1,10-phenanthroline, and ferrous acetate as catalyst precursors obtained by ball-milling and pyrolyzing in Ar at 950 °C and in NH_3_ at 1050 °C. The catalyst showed an approximately 35-fold increase in kinetic activity compared to previously reported non-precious metal catalysts (NPMC). In 2011, Proietti et al. [46] reported a ZIF-8-derived Fe/N/C catalyst that showed comparable performance to Pt-based catalyst with a power density of 0.75 W cm^−2^ at 0.6 V. At present, MOF-derived NPMC are widely used in various catalytic fields especially in oxygen reduction reaction [47,48,49]. MOF material is an ORR catalyst, since its abundant pores can promote the transportation of O_2_, and rich active metal sites can participate in redox reactions [50]. It can also be used as a precursor for the preparation of oxygen electrocatalysts, using transition metal ions as catalyst active centers to prepare NPMC [51,52], or using the volatility of low-boiling metals to prepare metal-free oxygen electrocatalysts [53,54]. MOF-derived NPMC for Zn–air batteries are introduced in this section, in which the structural characteristics of catalyst materials are mainly described, as well as the ORR and OER catalytic performance of the catalysts. After that, MOF-derived metal-free catalysts for Zn–air battery are introduced. 

### 3.1. MOF-Derived Single Metal Catalyst

#### 3.1.1. MOF-Derived Co-Based Catalysts

MOF composed of metallic cobalt and nitrogen-containing ligands is an ideal precursor for the preparation of Co-N coordination porous carbon electrocatalysts, due to its controllable structure, high porosity, large specific surface area, uniform dispersion of Co and N atoms, etc. [55,56,57]. Zeolitic imidazole framework (ZIF) is a subclass of MOFs materials, which has excellent flexibility and high specific surface area, are great precursors for multifunctional materials. Ma et al. [58] reported for the first time that Co zeolite imidazole skeleton was transformed into Co-N_x_ porous carbon material with ORR activity. Mu et al. [59] used ZIFs nanocrystals with high specific surface area as precursors for the first time, and converted them into Co-N_x_/C nanorods through pyrolysis. The Co-N_x_-C nanorods have outstanding ORR and OER electrocatalytic activity and stability, which are comparable to Pt/C and IrO_2_. Notably, this bifunctional electrocatalyst also confers the Zn–air battery with excellent performance and high energy density. This provides a feasible method for designing efficient ORR/OER electrocatalysts. A new strategy by adding another metal with low boiling point (such as Zn, boiling point 907 °C) into the precursor was adopted to improve the catalytic activity. Duan et al. [60] synthesized novel Co-MOF,O-doped carbon material by pyrolyzing catalyst precursor containing Zn, Co-doped glucosamine, and ZIF-8. Zn species could be evaporated and inhibited the aggregation of Co atoms during pyrolysis. The prepared Co-MOF-800 not only has the same ORR activity as commercial Pt/C, but also has amazing hydrogen evolution reaction (HER) catalytic activity, showing great application potential in water cracking and Zn–air battery.

Hollow nanostructures are ideal materials to control the local chemical environment of electrocatalyst reaction due to their characteristics of low density, thin shells, and high permeability. The internal cavity of the nanoparticle catalyst not only provides an additional three-phase interface to accelerate the reduction and evolution of O_2_, but also facilitates the diffusion of aggregated reactants on the catalyst [61,62]. Liu et al. [63] obtained a core–shell structure ZIF-8@ZIF-67 crystals by epitaxially growing ZIF-67 on prefabricated ZIF-8 nanoparticles. The double-shell hybrid nanocage NC@Co-NGC DSNCs with Co-N doped graphite carbon (Co-NGC) outer shell and N-doped microporous carbon (NC) inner shell were formed after high-temperature pyrolysis and pickling (Figure 4a). The ZIF-67-derived Co-NGC outer shell had high ORR catalytic activity with a stable structure and excellent conductivity, while the ZIF-8-derived NC inner shell promoted diffusion kinetics with a nanostructured hollow skeleton. The nanomaterial exhibited better electrocatalytic performance than Pt/C and RuO_2_ using as a bifunction electrocatalyst for ORR/OER (Figure 4b). The simulation reveals that the intermediate (OOH*) has strong and favorable adsorption on the non-coordinating hollow-site C atoms relative to Co lattice, which may be accountable for the excellent bifunctional catalytic performance of the catalyst. Zhou et al. [64] prepared Co/CoO@NSC bifunctional electrocatalyst using Zn-MOF@Co-MOF as a template. The composite material with the interconnected porous structure exhibited excellent ORR/OER catalytic activity and stability. It is expected to become a high-efficiency air electrode catalyst for Zn–air battery. The hollow nanostructures were formed under high temperature. This hollow structure could improve the diffusion kinetics of the catalyst during ORR/OER. So, the catalyst showed excellent ORR/OER catalytic activity and high power density when it was used in the Zn–air battery.

The active Co_3_O_4_ nanoparticles closely connected to the carbon skeleton can obtain better electrochemical performance due to the advantages of high conductivity, excellent dispersibility, and rich porous structure [65]. The pyrolysis of ZIF-67 is a facile and effective strategy to obtain porous Co_3_O_4_ nanoparticles [65,66]. Rich oxygen vacancies and tetrahedral Co^2+^ can be found in these Co_3_O_4_ nanoparticles. Because of the synergistic effect between CoO_x_ species and nitrogen-doped carbon, this kind of catalyst exhibit excellent ORR/OER performance and good cycling performance in Zn–air battery. It provided a new way to prepare highly dispersed transition metal/metal oxide nanoparticles with multi-level structure, which improves a simple strategy for the development of high-performance non-noble metal bifunctional catalysts. Carbon (2.55) has a similar electronegativity to gold (2.54) and has good electrical conductivity. Combining carbon materials with MOFs is an effective way to improve the electrical conductivity of MOF-derived catalysts. Benzimidazole is an aromatic precursor, which has been proved to effectively produce graphite carbon with higher catalytic activity [67,68]. Zeolite imidazole framework-9 (ZIF-9) is a sodalite topology structure with hexagonal symmetry constructed by co-angular tetrahedral CoN_4_ units, in which the coordination bond between Co^2+^ and benzimidazole (PhIm) anion is the most stable N coordination ligands [69]. Li et al. [70] prepared ZIF-9-derived catalysts by introducing MWCNTs to overcome the poor electronic conductivity of the catalysts. The prepared hybrid product Co_3_O_4_@C-MWCNTs showed excellent OER/ORR bifunctional catalytic activity since the uniform distribution of Co_3_O_4_ nanoparticles, high nitrogen doping amount, large specific surface area, and clear mesoporous structure in catalysts.

The single atom catalyst (SAC) is a supported catalyst in which the metal is on a solid support in the form of a single atom. In traditional supported catalysts, only a few metal-active components play a role in the catalytic process. In order to improve the catalytic activity, only the loading of noble metals can be increased, which is too costly and is not conducive to large-scale utilization. Porous Co-N-C materials can be prepared by pyrolysis of MOF precursors at high temperature and the Co-N_x_ active center can be formed directly from the original Co-N coordination bond. However, severe agglomeration of Co atoms in the catalyst during high-temperature pyrolysis affects the formation of Co-N_x_ active sites and reduces the activity of the catalyst. Zang et al. [71] reported a Co single-atom catalyst in a nitrogen-doped porous carbon nanosheet array prepared by simple carbonization and acid leaching with MOF material as the precursor Co-N_x_ coordination and Co aggregates (Co-Co coordination) were formed during high-temperature carbonization (Figure 5a). A large number of Co clusters were dissolved and part of the Co-Co bonds were broken while Co single atoms coordinated with N were retained in the N-doped carbon flakes after acid washing (Figure 5b–d). The results of HAADF-STEM coupled with EELS showed that the coexistence of Co and N in the form of Co-N_x_. The Co single-atom catalyst exhibited high ORR and OER performance.

In conclusion, Co-based catalysts can be obtained by direct pyrolysis of Co-MOF, Co-doped ZIF materials, and ZIF-8@ZIF-67 core–shell materials. The catalytic performance of Co-based catalysts can also be improved by designing single-atom catalysts and carbon materials composites. The Co-N_4_ appears to be the active center of Co-based catalysts [72,73]. Because the adsorption energy of O_2_ on CoN_4_ is low, the ORR could proceed with 2e^−^ pathway. Thus, Xing and co-workers [74] proposed that Co_2_N_5_ active sites are active for the ORR. They considered that the Co-N path located at 2.12 Å in Co K-edge spectra corresponds to the structure of the bimetallic atom (Co_2_N_x_). The density functional theory (DFT) calculations revealed that the novel binuclear site exhibits considerably reduced thermodynamic energy barrier towards the ORR when compared to CoN_4_, thus contributing to enhanced intrinsic activity.

#### 3.1.2. MOF-Derived Fe-Based Catalysts

Fe-N-C catalysts have the highest ORR activity among M-N-C (M = transition metal) [75,76,77]. However, there are few reports on the direct pyrolysis of Fe-MOF materials for oxygen redox reaction. Lefèvre and his colleagues [45,46,78,79] prepared Fe-N-C catalyst by ball milling of carbon supports mixed with iron salts showed excellent ORR activity, the iron-based catalyst prepared by ZIF-8 as support has the same catalytic potential as the Pt-based catalyst, which is of great significance for the preparation of ZIFs-derived NPMC. Lai et al. [80] reported a Fe-N/C electrocatalyst with controllable structure obtained by pyrolysis and acid leaching using Fe-mIm nanoclusters(NCs)(guest)@zeolite imidazole frame-8 (ZIF-8)(host) as catalyst precursor (Figure 6a). A 2–5 coordinated Fe-N_x_ structure was formed during pyrolysis with different Fe-mIm contents in ZIF precursors (Figure 6b). Electrochemical tests and DFT calculations showed that the five-coordinated Fe-N_x_ center significantly improved the catalytic activity and selectivity for ORR in acidic medium, by reducing the reaction energy barrier and the adsorption energy of the intermediate OH. This synthesis strategy provides an effective way to construct the Fe-N_x_ active center with clear structure, and makes it possible to further reveal the relationship between catalysts structure and electrocatalytic performance. M-N_4_ structures widely existing in many metal macrocycles such as metal phthalocyanine (MPc) and metalloporphyrin (MP) have been proved to be helpful for ORR [81,82,83]. Cheng et al. [84] synthesized iron-polyphthoalocyanine FePPc MOFs on carbon black matrix (FePPc@CB) by melt polymerization strategy. FePPc molecules can be adsorbed on the carbon matrix to promote the electron transfer process and stabilize the catalytic system through strong non-covalent π-π interactions. FePPc@CB exhibited excellent ORR/OER activities as bifunctional electrocatalyst, owing to the abundant free electrons and M-N_4_ catalytic centers on atoms in the macrocyclic structure of FePPc@CB. The carbon matrix with high electrical conductivity allows efficient electron transportation during the electrochemical ORR process, which is beneficial to enhance the intrinsic activity of the catalyst. The M-N_4_ provided by this metallo-macrocyclic molecule has abundant active centers. Consequently, this work will open new avenues to design M-N_4_ materials with good atomic dispersion and efficient catalytic performance.

Because of the uneven distribution of Fe species in the precursor, heterogeneous structures and metal nanoparticles (NPs) were obtained by annealing Fe-doped MOF-derived catalysts [85,86]. The active metal NPs are covered by a thick carbon layer and hinder the catalytic process. Thus, it is necessary to find an efficient way to prepare catalysts with uniform Fe distribution. Besides, the carbon materials such as CNTs and graphene show high specific surface area and good electrical conductivity. So, building MOFs/carbon materials composite is an effective way to improve catalytic performance [87,88]. Zhao et al. [89] reported a Fe_3_C@NCNT/NPC with a hybrid structure of Fe_3_C nanorods encapsulated N-doped CNT grown on porous carbon sheet by simply annealing a Fe-based MOF (MIL-88B) loaded with melamine. Fe_3_C@NCNT/NPC exhibited excellent ORR/OER performance due to the high porosity and high electrical conductivity in NCNT/NPC hybrid material and high density of Fe-N active in catalyst. This work provides a new strategy for in-situ construction of metal carbide nanorods encapsulated N-doped carbon nanotubes with different types of carbon, and a feasible route for the design of multifunctional NPMC. Xiong et al. [51] prepared iron-nitrogen co-doped porous carbon (Fe-N-HPC) by pyrolysis of MWCNTs@PDA@Zn-Fe-ZIF (Figure 7a). ZIF containing aromatic imidazole ligands are rich in nitrogen and carbon. Multi-walled carbon nanotubes (MWCNTs) are excellent graphite carbon conductive skeleton materials. Polydopamine (PDA) serves as a linker and nitrogen source. The surface of MWCNTs@PDA is wrapped by rhombic dodecahedral polyhedral Zn-Fe-ZIF crystals with a diameter of 70 nm (Figure 7b,c). Zn-Fe-ZIF is obtained by using ZIF-8 molecular sieve as raw material and performing step-by-step ion replacement with Zn and Fe. The formation of porous carbon sheets generates larger specific surface area and efficient active centers (Figure 7d,e), which is more favorable for the electrocatalytic performance of ORR and OER. The as-prepared Fe-N-HPC catalyst combines the advantages of ZIFs (tunablity of structures and functions, larger specific surface area, higher porosity) and MWCNTs (good electrical conductivity), with efficient oxygen electrocatalytic performance, long-term stability, and excellent methanol resistance. In conclusion, MOF/CNT composites with efficient electrocatalytic performance were prepared. This provides a feasible way to prepare promising electrocatalysts.

#### 3.1.3. MOF-Derived Other Catalysts

Among M-N-C catalysts, although Fe-N-C catalysts have outstanding activity, the catalyst still faces great challenge—the severe Fenton effect, that is, the reaction of dissolved Fe^2+^ with the hydrogen peroxide generated by the ORR process will generate free radicals, which free radicals can lead to a decrease in catalyst stability [90]. The research shows that the active site Mn-N_x_ formed by Mn salt in acid is relatively stable, and the Fenton effect is one thousandth of that of Fe, which is negligible. In 2018, Li et al. [91] prepared an atomically dispersed Mn-N-C catalyst for the first time by two-step doping and adsorption (Figure 8a). In acidic medium, the half-wave potential (0.80 V vs. RHE) of Mn-N-C catalyst is close to that of Fe-N-C catalyst, and has outstanding ORR stability (Figure 8b–d). Computational analysis indicated that the MnN_4_ structure might exist in the catalyst and was considered to be an active site favorable for ORR. This work provides a new idea for the development of durable and highly active Fe-free non-precious metal catalysts. Han et al. [92] prepared a Mn-SAS catalyst (Mn-SAS/CN) with Mn-N_4_ structure by thermal activation strategy. The test results show that Mn^L+^-N_4_ is the active center in the ORR process, and the atomically dispersed Mn^L+^-N_4_ sites can facilitate the transfer of electrons to ^*^OH species. Mn-SAS catalysts exhibit high power density and excellent durability when assembled into a Zn–air battery. This provides a promising choice for the design of non-Fe single-atom catalysts, and also provides ideas for further understanding the ORR active center of Mn-N_x_ catalysts. Mn ions in manganese oxides exist in mixed valences, which can promote electrolyte diffusion and electron transfer during ORR process [93]. Najam et al. [94] prepared a structurally interesting Mn_3_O_4_@NCP catalyst by directional growth of manganese oxide (Mn_3_O_4_) quasi-nanocubes on nitrogen-doped mesoporous carbon polyhedron formed by zeolitic imidazole framework (ZIF-8). In this novel hybrid structure, NCP provides high specific surface area and porous structure, which improves the electronic conductivity of Mn_3_O_4_@NCP, and Mn_3_O_4_ plays the role of improving the catalytic active center. Further studies show that Mn_3_O_4_@NCP exhibits 4e ORR mechanism, less production of hydrogen peroxide, strong methanol tolerance, and stability. This method for the synthesis of mesoporous carbon is called nanoengineering directed growth (NEDG) method, which can be used for surface/interface modification, and also for the synthesis of new nanomaterials for energy conversion and storage. Introducing the conductive material graphene into the MOF can improve the conductivity of the catalyst [87]. Wahab et al. [95] utilize GO template assist synthesis of MnBDC MOF@rGO nanocomposites with different GO contents. GO is reduced to rGO by continuous thermal reduction, and strong coordination solvent molecules that may reduce the quality of the catalyst are removed to activate the nanocomposites. The template directed growth, tunable porosity, and novel structure enable the MnBDC@75% rGO catalyst to exhibit excellent ORR/OER bifunctional catalytic activity, and its ORR activity is comparable to that of commercial Pt/C catalysts. Nanocomposites synthesized by this method can contain desired structural features such as mesoporous surfaces, controllable growth, and tunable functionality.

Compared with Fe-N_4_, Co-N_4_, and Mn-N_4_, the Cu-N_4_ coordination configuration has smaller oxygen molecule adsorption energy, resulting in insufficient ORR activity [96]. This may be because the unique molecular structure of Cu-N_4_ provides less d-orbitals and more steric hindrance for binding oxygen molecules [97,98]. However, the in-depth study of Cu-N-C catalysts is of great significance to understanding the electrocatalytic mechanism of M-N-C catalysts. Lai et al. [99] successfully prepared a new type of Cu-N/C ORR electrocatalyst by using metal doping-induced synthesis strategy to control the doping of Cu^2+^ in situ in ZIF-8. The obtained Cu-N/C catalyst can maintain the polyhedron morphology of ZIF-8 (Figure 9A–C), with high specific surface area (1182 m^2^ g^−1^), refined hierarchical pore structure and high surface nitrogen content (11.05 at%). After acid leaching and pyrolysis, the catalyst can still maintain the original polyhedron morphology (Figure 9D–H). The optimized 25% Cu-N/C catalyst possesses a high ORR activity and stability in 0.1 M KOH solution, as well as excellent performance in a Zn–air battery (Figure 9I–L). The metal doping induced synthesis strategy of MOF shows significant advantages in adjusting the metal hybrid structure, and this work also provides a new idea for the preparation of MOF material derived M-N-C catalyst.

In this section, MOF-derived single NPMCs are introduced (Table 1), which provide a feasible idea for the preparation of catalysts with ORR/OER bifunctional catalytic activity. Defining the accurate active site will facilitate optimal the framework of M-N-C catalysts. The unstable structure of the M-N-C catalyst leads to insufficient stability. Here, we list some countermeasures: the utilization rate of catalyst active sites can be improved by preparing single-atom catalysts and using the strategy of low boiling point metal replacement; the conductivity of the catalyst can be improved by introducing conductive materials; through pyrolysis and acid leaching treatment, excess metal species, and strong coordination, solvent molecules that may reduce the activity of the catalyst can be removed, so as to prevent the aggregation of metal species and activate the catalyst; using the similarity between the two ZIF materials, the core–shell materials with novel structure are prepared, which provides a new way for the design of ORR/OER bifunctional NPMC; and the hollow structure can improve the stability of the catalyst. Most importantly, this section reviews the preparation methods and processing methods of MOF-derived M-N-C (M = Co, Fe, Mn, Cu) bifunctional catalysts, and further studies and improves the ORR/OER bifunctional catalytic performance of M-N-C catalysts, which is conducive to a more extensive and in-depth understanding of the point catalysis mechanism of the TM-N-C system.

### 3.2. MOF-Derived Bimetallic Catalyst

Compared with single metal NPs, bimetallic alloy NPs have higher catalytic efficiency due to the strong synergy between different kinds of metals. Bimetallic NPMC such as Cu, Mo, Mn, Fe, Ni, W, Zn, Cr, and Co are frequently used as NPMC [33,100,101,102,103,104]. Incorporating another metal to transition metal catalysts is considered to be an effective method to improve their catalytic activity [105,106]. However, there are still great difficulties to obtain high-performance bimetallic catalysts. On one hand, simple mixing of the metal precursors may cause a multiphase structure. On the other hand, the limited active sites in bimetallic catalysts are mainly caused by the sintering and agglomeration during the metal deposition process. MOFs, as precursors of bimetallic NPs, help to solve these problems due to their elemental advantages. Here, the synthesis methods, structural characteristics, and applications in Zn–air battery of MOF-derived bimetallic catalysts, such as FeCo, FeNi, and CoNi, are briefly introduced.

#### 3.2.1. MOF-Derived Iron-Containing Bimetallic Catalysts

Bimetallic doping ZIF-8 is a feasible method to prepare bimetallic catalysts as the zinc ions in ZIF-8 can be partially evaporated and replaced by other transition metals. Li et al. [107] prepared a layered porous structure composed of nanotubes, nano-blocks, and encapsulated FeCo alloy NPs with the high specific surface area through the hybridization of Fe (II) doped ZIF-8 (Fe-ZIF) and cobalt acetylacetonate (Co(acac)_3_) (Figure 10a), which greatly promoted mass transfer and electron transfer. The FeCo-NC-850 catalyst shows excellent ORR activity and stability in alkaline medium (Figure 10b–e), due to the high specific surface area, the synergistic effect of the FeCo alloy, and the unique structure of the bimetallic-N active center, which is a promising bifunctional oxygen catalyst. Fe-doped ZIF-67 is a feasible strategy for MOF-derived FeCo bimetallic catalysts. Zhang et al. [52] prepared Fe-Phth-CMP@ZIF-67 composites with Fe-phthalocyanine based conjugated microporous polymer (Fe-Phth-CMP)-coated metal–organic framework ZIF-67 by MOF template-assisted method (Figure 11a). The composite has controllable morphology and an adjustable Fe/Co molar ratio. After carbonization, it can be further converted into N-doped porous carbon (P_m_Z_n_-900) with adjustable N content and embedded in highly dispersed FeCo alloy and Fe/Co-N active center. The optimized catalyst (P_2_Z_3_-900) has a hierarchical pore structure (Figure 11b–d), which can significantly improve the mass transfer efficiency, increase the exposure of active sites, and exhibit excellent ORR/OER bifunctional catalytic activity under alkaline conditions. The current strategy provides a feasible method for the preparation of porous carbon with controllable structural morphology and elemental composition, so as to adjust the catalyst performance to achieve efficient catalysis.

ZIF-8 has high specific surface area and high N content. ZIF-67 has a high degree of graphitization and abundant Co-N-C active sites. ZIF-8/ZIF-67 combines the advantages of the two ZIF materials (high specific surface area, high nitrogen content, etc.), and the strong coupling between the two ZIF materials also improves the stability of the catalyst. The FeCo bimetallic catalyst prepared by Fe-doped ZIF-8/ZIF-67 has more active sites and higher stability by the substitution of low-boiling metals. Thence, Luo et al. [108] atomically dispersed Fe and Co doped 3D nitrogen-doped carbon sheets (A-FeCo@NCNs) were prepared by facile and effective approach with a modified zeolite imidazole framework (SiO_2_@Fe-ZIF-8/67). A-FeCo@NCNs catalyst exhibits bifunctional catalytic performance superior to commercial Pt/C and IrO_2_ catalysts because of its high specific surface area (809.23 m^2^ g^−1^), tunable microporous structure, abundant bimetallic monatomic active centers (FeN_4_, CoN_4_, and N_4_Fe-CoN_4_), and synergistic coupling between metals. Zn–air battery with A-FeCo@NCNs as air cathode has high power density and specific capacity. Duan et al. [109] prepared a novel layered Fe,Co@N-C bifunctional oxygen electrocatalyst (FeCo-N-C-T) using Fe^3+^ and glucosamine-coated ZIF-8/67 as supports (Figure 12a). The test results show that FeCo-N-C-700 is a highly active bifunctional electrocatalyst with better ORR and OER activities than commercial Pt/C and RuO_2_ catalysts, respectively (Figure 12b). The smaller polarization indicated better performance at high current density for the FeCo-N-C-700-based ZABs than the corresponding battery based on an air-cathode made from Pt/C + RuO_2_ as a bifunctional catalyst for ORR and OER (Figure 11c). FeCo-N-C-700-based rechargeable battery showed excellent performance and stability (Figure 12d,e). DFT simulation show that Fe and Co in FeCo bimetallic catalysts affect the electronic structure, which has a synergistic effect on improving catalytic activity. Considering the weak electron transport ability of MOF-derived carbon materials, Fang et al. [110] mixed reduced graphene oxide (rGO) with MOF-derived nitrogen-doped CoC_x_/FeCo@C core–shell structure. The special heterojunction structure formed by the interconnection of the core–shell CoC_x_/FeCo@C and rGO sheets provides a large specific surface area and stable active sites for ORR and OER. N-doped CoCx/FeCo@C/rGO catalyst shows excellent bifunctional catalytic performance. This depends on several things: (i) Atomically dispersed Fe and Co can provide more active sites. (ii) Rich N doping can modulate the electronic structure. (iii) The core–shell structure can increase the number of accessible active sites.

Currently, most of the MOF-derived Fe-containing bimetallic catalysts focused on FeCo bifunctional electrocatalyst. Other Fe-containing bimetallic ORR/OER catalysts are mainly supported by carbon materials such as graphene, carbon nanotubes, and g-C_3_N_4_, such as FeMn-based bifunctional electrocatalyst [111,112,113], FeNi-based bifunctional electrocatalyst [114,115,116,117,118], and FeCu-based bifunctional electrocatalyst [119]. There are few reports on other Fe-containing bimetallic catalysts derived from MOF for ORR/OER.

The carbon matrix derived from Zn-ZIF has rich N doping content and uniform porosity. Yao et al. [120] embedded Fe, Ni-based nanoparticles into the N-doped carbon sheets prepared by Zn-ZIF using low boiling point metal replacement. The prepared FeNi-NCS-2 catalyst has a large specific surface area, a hierarchical sheet-like porous structure, and a large number of transport channels provided by carbon nanotubes. The Zn–air battery with FeNi-NCS-2 as the air cathode shows large open-circuit voltage, high power density, and excellent cycling stability. This work may drive a new insight to design efficient bifunctional electrocatalysts through an economic and environmental way for high performance rechargeable Zn–air battery. Wu et al. [121] prepared FeNi alloy catalysts encapsulated within N-doped carbon (FeNi@NCNT) by carbonizing Fe-doped ZIF-8 coated with PDA/Ni^2+^ complex. It is worth noting that a certain proportion of Fe/Ni can catalyze the growth of one-dimensional bamboo-like carbon nanotubes, forming a conductive network to facilitate the species transport during ORR/OER. The composite exhibits excellent stability and high ORR/OER activity under alkaline conditions, providing a new perspective for exploring advanced iron-containing bimetallic catalysts. A MOF-derived FeNi co-doped catalyst (B-FeNi-N/C-1000) was synthesized by Cui et al. [122] using a binary coordination strategy. B-FeNi-N/C-1000 has bifunctional catalytic performance due to its large specific surface area, high pyridine N and graphitic N content, and suitable hierarchical pore structure, its ORR performance is superior to that of Pt/C, and OER performance is superior to that of IrO_2_/C. Moreover, the practical application in a Zn–air battery is also better than that of Pt/C + IrO_2_/C. This strategy can be further applied to the synthesis of catalysts with appropriate hierarchical pore structure, which can effectively catalyze ORR and OER.

In conclusion, the Fe-containing bimetallic catalyst derived from MOF has a hierarchical porous structure and abundant active sites, showing better bifunctional catalytic performance than Pt/C + RuO_2_, which may be due to the synergistic coupling between bimetallic materials, the unique structural advantages of MOF materials (rich N content, adjustable pore structure, large specific surface area) and novel synthesis strategies. Designing Fe-containing bimetallic ORR/OER catalysts is a feasible way to replace Pt/C + RuO_2_ catalysts. In this subsection, novel synthesis methods and processing methods for MOF-derived Fe-containing bimetallic catalysts are reviewed, which bring new hope for the development of transition metal catalysts to replace noble metal catalysts.

#### 3.2.2. MOF-Derived Non-Iron Bimetallic Catalysts

MOF-derived Fe-containing bimetallic catalysts have excellent activity. However, Fe-doped carbon-based catalysts show poor stability in the practical battery because of serious Fenton effect. Therefore, it is necessary to study non-Fe bimetallic catalysts.

Ultra-thin MOF two-dimensional nanosheets (UMOFNs) with nano-thickness can allow rapid mass transfer and excellent electron transfer, large catalytic surface, and coordination of unsaturated metal sites to ensure high catalytic activity; it has a unique surface atomic structure that is easy to identify and adjust [123]. Therefore, Huang et al. [124] developed a NiCo-MOF oxygen electrocatalyst supported on nickel foam by the growth–pyrolysis–regrowth strategy, which has a good two-dimensional MOF/MOF derivative coupling array and high ORR/OER bifunctional electrocatalyst activity. In addition to ultra-thin two-dimensional nanosheets, researchers also hybridize or composite MOFs materials with conductive carriers, such as reduced graphene oxide (rGO) and CNTs [125]. Zheng et al. [126] reported the synthesis of bimetallic CoNi-MOF nanosheet/reduced graphene oxide (rGO) hybrid electrocatalyst. With the assistance of surfactants, CoNi-MOF nanosheets were grown in situ on rGO (Figure 13a). CoNi-MOF/rGO has a fine synergistic effect and more exposed active sites, and exhibits excellent bifunctional catalytic activity and high durability in alkaline electrolyte (Figure 13b,c). Zn–air battery with CoNi-MOF/rGO as air cathode has high peak power density and energy density, and excellent stability (Figure 13d,e). Through epitaxial growth of Co-MOF on nickel complexes, Li et al. [127] formed Co/Ni atomic double sites in N-doped porous carbon Janus skeleton. The synthesized Co/Ni-N-C catalyst has two different topological structures, forming a large number of uniform Co/Ni atomic centers, which is conducive to the improvement of electrocatalytic performance. DFT proved that the presence of an Ni-N bond improved the electronic activity of N atom, which was the main reason for the improvement of the bifunctional catalytic performance of the catalyst ORR/OER. This work opens up a new way for the development of efficient multiatomic Janus electrocatalysts for energy conversion.

At present, the research on non-Fe bimetallic catalysts derived from MOF has only just begun. Most CoMn [128,129,130,131] and CoNi-based catalysts [132,133,134,135] are still supported by carbon materials such as graphene and carbon nanosheets. Considering the unique structure and abundant composition of MOF materials, designing MOF-derived non-Fe bimetallic catalysts is a feasible strategy for the development of NPMC, which will accelerate the process of replacing noble metal oxygen electrocatalysts with NPMC. Table 2 briefly summarizes the ORR/OER performance of MOF-based bimetallic catalysts.

## 4. MOF-Derived Non-Metallic Catalysts

N, B, P, S, and other heteroatoms doped carbon materials can effectively change their electronic properties, and generate catalytic active sites by inducing charge and spin density on the carbon atoms near the dopant [136,137,138,139]. N atom has the characteristics of high electronegativity, oxidation stability, and similar size to C atom, which is the most widely studied doped atom among all heteroatoms. N-doped carbon material has been proved to be an effective metal-free electrocatalyst [140,141]. Theoretical simulation [142] and experimental results [143] show that the excellent catalytic activity of NC material for ORR stems from the substitution of C atoms by N atoms in carbon matrix sp^2^ lattice, which destroys the electrical neutrality of adjacent C atoms and produces a positive charge potential conducive to O_2_ adsorption or cracking. Therefore, the type and content of doped N are crucial for the electrocatalysis reaction. In addition, the catalytic performance of NC materials is also strongly dependent on the specific surface area and porosity of the carbon matrix [144,145]. MOF has a variety of available metal centers, abundant heteroatom-doped organic ligands and porous structures, and the preparation of porous metal-free nitrogen-doped carbon catalysts from MOF materials have been widely studied. Qian et al. [146] prepared a B-N double-doped metal-free porous carbon material for the first time by pyrolysis of Zn-MOF (MC-BIF-1S) in a mixed H_2_-Ar atmosphere. In the crystalline MOF precursor, N and B are uniformly distributed on the carbon material, and pyrolysis in a hydrogen-containing atmosphere can greatly increase the specific surface area of the carbon material. The high porosity of these carbon materials and the introduction of N and B can effectively improve their ORR/OER performance. Li et al. [147] treated ZIF-8 with butyl methylphosphonate to obtain highly branched N,P co-doped carbon nanotube clusters (NPCTCs) (Figure 14a). NPCTC-850 has better bifunctional catalytic performance (Figure 14b), better ORR performance than Pt/C, and better OER performance than IrO_2_, because the ultrathin structure is conducive to the full utilization of the catalytic active centers inside and outside the carbon nanotube cluster (Figure 14c–h), the nanotube cluster itself has excellent bifunctional catalytic activity, and N,P co-doping enhances the activity of nanotube clusters. The Zn–air battery based on the NPCTC-850 air cathode exhibits a peak power density of 74 mW cm^−2^ and an energy density of 896 Wh kg_Zn_^−1^, and has good charge–discharge cyclability. This work demonstrates that chemical modification can improve the electrochemical activity and mechanical strength of carbon materials, providing a facile strategy for MOF-derived metal-free catalysts.

Graphene oxide (GO) is the oxidative graphite, which is dispersed and can be used as the precursor of graphene after chemical thermal reduction [148]. GO films have excellent solubility and openness, and the surface is rich in oxygen-rich functional groups (such as -OH and -COOH). Functionalized graphene can be used as a structural guidance template to affect the crystal growth of MOF [87,148,149]. The conductivity and high specific surface area of graphene can give MOF new properties. The results show that ZIF nanocrystals can be controllably grown in situ on the GO surface to form a sandwich-like structure ZIFs@GO composite [150,151,152]. After high-temperature pyrolysis, N-doped porous carbon@graphene composite can be obtained. Liu et al. [54] obtained zeolite imidazole framework (ZIF-8)@graphene oxide composite (ZIF-8@GO) by in-situ controllable growth of ZIF-8 nanocrystals on both sides of GO sheets. A sandwich structure of nitrogen-containing porous carbon@graphene (N-PC@G) composite was prepared by pyrolysis, and the zinc was etched (Figure 15a). The prepared N-PC@G-0.02 catalyst (with a GO content of 0.02 g in the reaction precursor) exhibits comparable ORR/OER bifunctional catalytic performance (Figure 15c,d). It could be attributed to synergistic effect between the N-doped porous carbon and N-doped graphene producing more catalytically active centers, while the high specific surface area facilitates the exposure of catalytically active centers, the porous structure (mesoporous and micropores) (Figure 15b) improves the mass transfer performance, and the high graphite and the degree of chemistry favor electron transfer. This provides an ideal strategy for designing sandwiched nitrogen-containing porous carbon@graphene hybrid materials with large specific surface areas and excellent electron transfer performance as high-performance ORR/OER bifunctional electrocatalyst.

The above strategies of heteroatom doping carbon materials and ZIF composite graphene oxide are feasible ways to prepare metal-free bifunctional electrocatalyst. At present, there are a few reports on MOF-derived metal-free bifunctional electrocatalyst (Table 3). Considering the structural advantages and rich composition of MOF materials, it is very important to design MOF-derived metal-free ORR/OER bifunctional electrocatalyst by using the volatility of low boiling point metals. This provides a new way for the development of metal-free catalysts and a new idea for the preparation of bifunctional electrocatalyst with ORR/OER performance.

## 5. Summary and Prospect

The energy problem is becoming more and more serious. Zn–air battery, as a kind of electrochemical energy conversion and storage device with rich resources and low price, has been widely studied in recent years. However, its large-scale application is still limited by the slow kinetics of ORR and OER in the air cathode during discharge and charge processes. Noble metal catalysts for air cathodes, such as Pt/C catalysts, RuO_2_ and IrO_2_, are single function catalysts with scarce resources, high prices, and poor stability. For the improved performance and durability of Zn–air battery, the exploration of efficient oxygen electrocatalysts is urgent and meaningful. This paper reviewed the research progress of transition metal electrocatalysts and metal-free electrocatalysts derived from porous organic crystal materials—MOFs.

The original MOFs can be used as electrocatalysts for oxygen redox reaction with poor catalytic due to their low conductivity which hinders the electron transfer in the system. In addition, the original MOFs have poor stability because their porous crystal structure is easy to collapse in acidic or alkaline solutions. However, the unique structural characteristics of MOFs provide advantages for the design of electrocatalysts as the most promising electrocatalysts precursor: metal ions are connected and separated by organic ligands, and multiple metal cations can be introduced by selecting organic ligands, providing more active sites for catalyst. MOFs derivatives usually retain the morphology of MOF precursors and have a high porous structure, providing abundant active surfaces for oxygen electrocatalytic reaction (ORR/OER). MOFs-derived NPMC are considered as one of the most promising candidates to replace platinum-based catalysts due to their low cost, abundant resources, and tunable catalytic activity.

The following strategies are proposed to improve the activity: (i) Combining MOFs materials with carbon materials such as CNTs, graphene, etc., can effectively solve the problems of poor conductivity of catalysts and agglomeration of metal ions. (ii) Doping MOFs materials with transition metal ions can provide more active sites for catalysts. (iii) Designing catalysts with hollow structure, and core–shell structure can improve the mass transfer and expose more active sites. (iv) Designing single-atom catalysts or using template-assisted growth methods are also effective ways to increase active sites. (v) MOFs-derived metal-free nitrogen-carbon (NC) materials with large specific surface areas and high electrical conductivity can provide more charge transport channels. Doping MOFs-derived metal-free NC materials with heteroatoms (N, P, S, B, etc.) or combining them with graphene is a feasible method to improve the catalytic performance of MOFs-derived metal-free NC materials. Heteroatom doping can modify the local electronic structure of metal-free catalysts and effectively improve the catalytic activity. Graphene combining can improve the degree of graphitization of the catalyst, which is beneficial to electron transfer. In addition, the synergistic effect between NC derived from MOFs and N-doped graphene can provide more active centers.

The future commercialization of Zn–air batteries requires us to start from the design of catalysts, electrodes, and batteries, and comprehensively consider performance, cost, and other issues. Design and construction of efficient electrode materials is always a priority. In summary, as emerging organic-inorganic hybrid porous material, MOFs have great development prospects in the design of unique bifunctional ORR/OER electrocatalysts. The main purpose of this review is to provide some feasible ideas for researchers to design environmentally friendly bifunctional ORR/OER electrocatalysts with high activity and selectivity, excellent stability, and conductivity. In addition, optimization of cell configuration, electrolyte, and operating conditions is required to improve the performance of Zn–air battery such as power density, energy density, and stability. For example, we can choose appropriate additives to overcome zinc passivation and dendrite formation; use filters based on chemical/physical absorption to reduce CO_2_ and prevent the deactivation of electrode materials caused by CO_2_ poisoning; and improve the utilization rate of electrolyte and prevent leakage. Although there are still many challenges, it is expected that the research results of MOFs materials can be used to solve the problem of slow air cathode kinetics of Zn–air battery soon.

## Figures and Tables

**Figure 1 materials-15-05837-f001:**
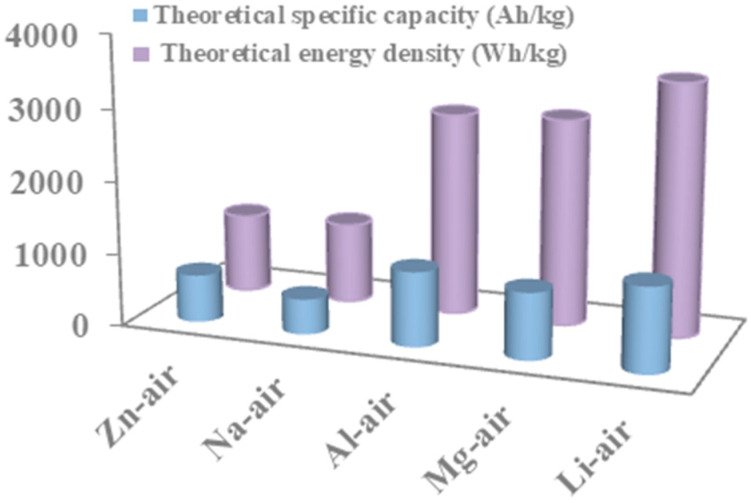
The theoretical specific capacity and theoretical energy density of various types of metal–air battery.

**Figure 2 materials-15-05837-f002:**
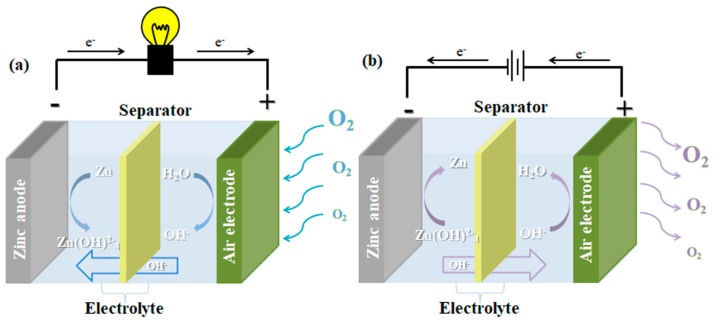
Schematic diagram of the working principle of Zn–air battery (**a**) discharge and (**b**) charge.

**Figure 3 materials-15-05837-f003:**
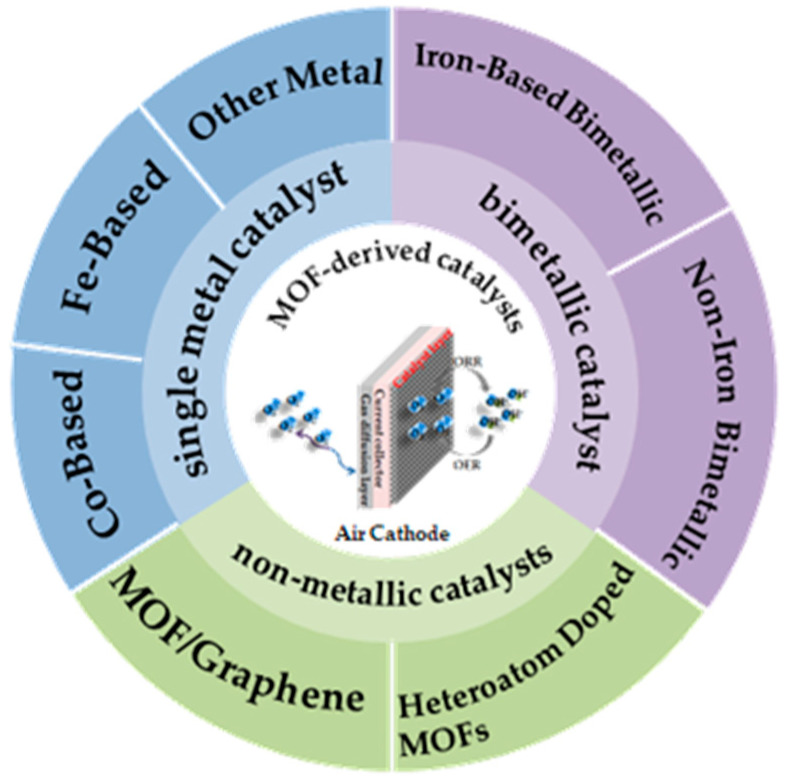
Classification diagram of MOF-derived catalysts for air cathode in Zn–air battery.

**Figure 4 materials-15-05837-f004:**
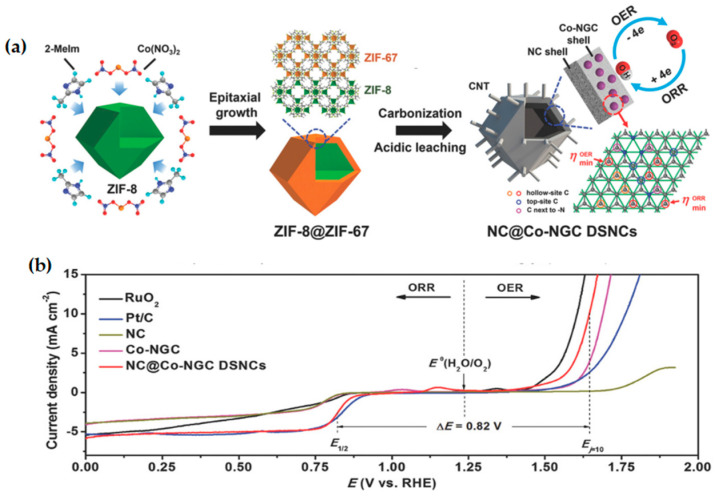
(**a**) Schematic illustration of the synthesis of NC@Co-NGC DSNCs; (**b**) I_R_-corrected polarization curves of NC, Co-NGC, NC@Co-NGC DSNC, Pt/C, and RuO_2_ catalysts, which are tested by using the three-electrode system in the full OER/ORR region in O_2_-saturated 0.1 M KOH solution. Copyright 2017 WILEY-VCH Verlag GmbH & Co. KGaA, Weinheim, Germany [63].

**Figure 5 materials-15-05837-f005:**
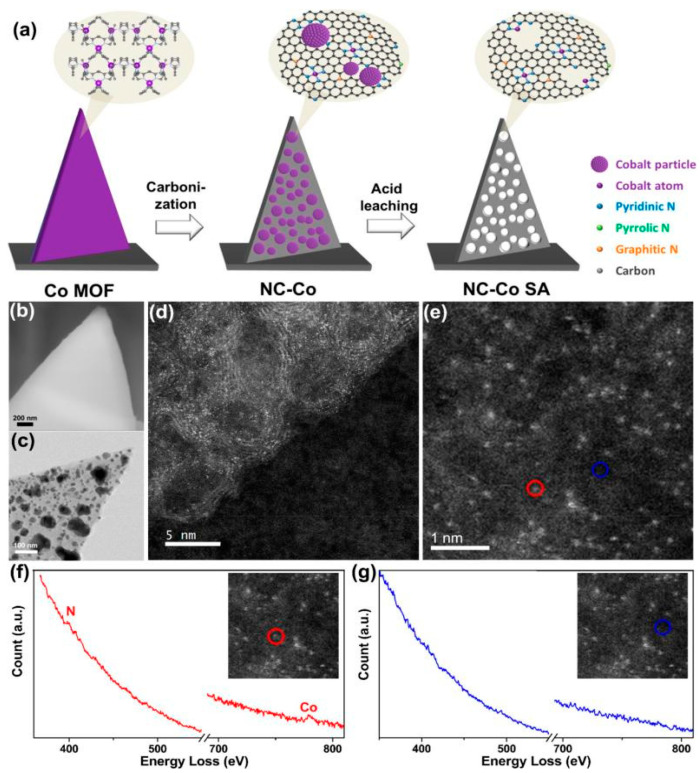
(**a**) Schematic diagram of the NC-Co SA fabrication process; (**b**) SEM image of Co MOF. (**c**) STEM image of NC-Co; (**d**) STEM image of NC-Co SA; (**e**) HAADF STEM image of Co atoms distributed across the nitrogen-doped carbonaceous support; (**f**) EEL spectra taken at the bright atom in the red circle in (**e**) showing Co and N edges; (**g**) EEL spectra taken at the dark support area in the blue circle in (**e**) showing neither. Copyright 2018, American Chemical Society [71].

**Figure 6 materials-15-05837-f006:**
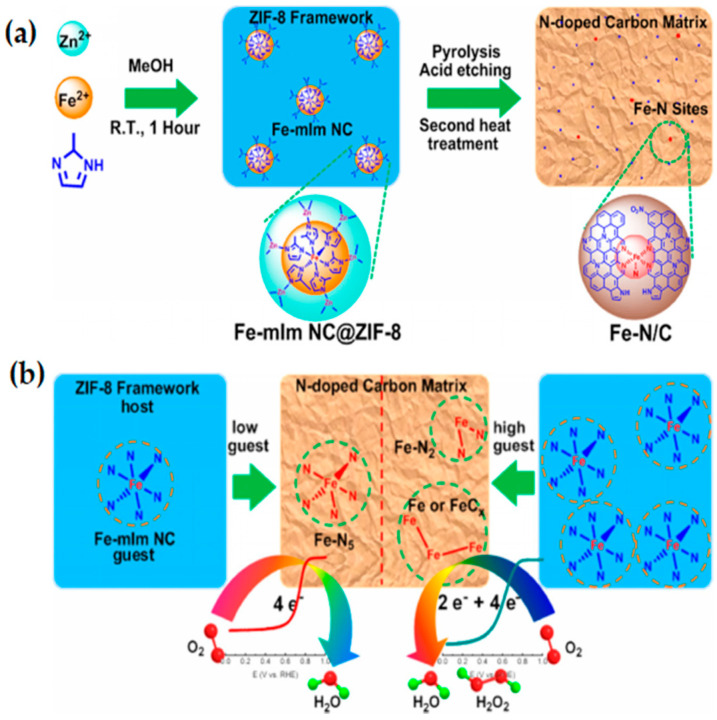
(**a**) Illustration of the host-guest chemistry strategy to fabricate MOF-derived Fe-N/C catalysts; (**b**) Schematic diagram of the formation of Fe-N_x_ structures with different coordination numbers. Copyright 2017, American Chemical Society [80].

**Figure 7 materials-15-05837-f007:**
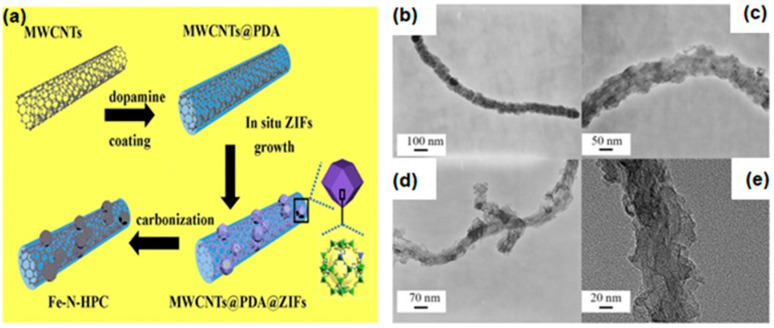
(**a**) Schematic illustration of the synthesis for Fe-N-HPC composites; TEM images of (**b**,**c**) MWCNTs@PDA@ZIFs and (**d**,**e**) Fe-N-HPC-900. Copyright Springer-Verlag GmbH Germany, part of Springer Nature 2019 [51].

**Figure 8 materials-15-05837-f008:**
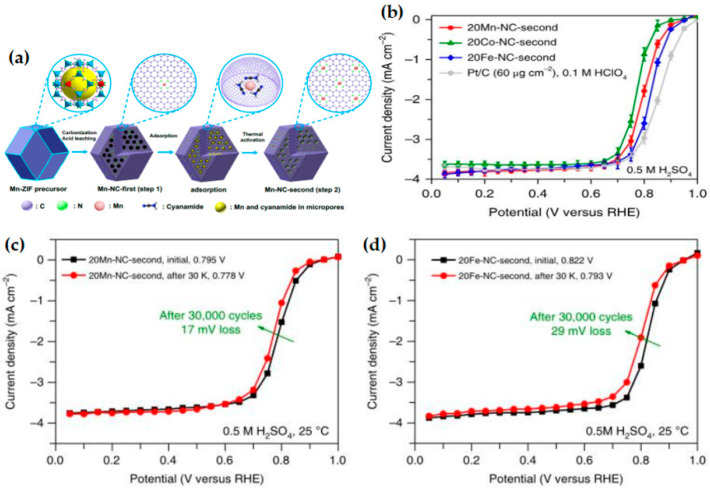
(**a**) Schematic of atomically dispersed Mn-N_4_ site catalyst synthesis; (**b**) comparison of catalytic activity of Fe-, Co- and Mn-N-C catalysts prepared from identical procedures, and steady-state ORR polarization plots before and after potential cycling stability tests for the 20 Mn-NC-second (**c**) and 20Fe-NC-second (**d**) catalysts. Copyright 2018, Nature Catalysis [91].

**Figure 9 materials-15-05837-f009:**
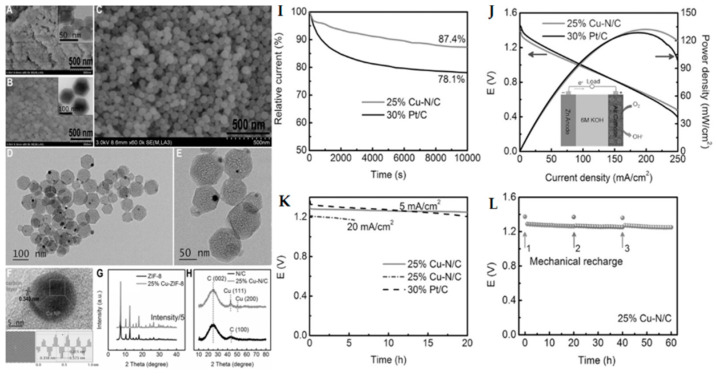
SEM images of (**A**) ZIF-8, (**B**) 25% Cu-ZIF-8, and (**C**) 25% Cu-N/C (inset in (**A**,**B**) corresponds with TEM image); (**D**–**F**) TEM images of 25% Cu-N/C. XRD patterns of (**G**) ZIF-8 and 25% Cu-ZIF-8, and (**H**) N/C and 25% Cu-N/C; (**I**) Normalized current–time responses of the ORR on 25% Cu-N/C, and 30 wt% Pt/C in O_2_-saturated 0.1 M KOH solution; (**J**) Polarization and power density curves of a Zn–air battery with 25% Cu-N/C and 30 wt% Pt/C as cathode catalysts, respectively (inset: scheme of the Zn–air battery); (**K**) Discharge curves of a Zn–air battery assembled from 25% Cu-N/C and 30 wt% Pt/C catalysts at a current density of 5 and 20 mA cm^−2^; (**L**) “Mechanical recharging” the Zn–air battery with 25% Cu-N/C as cathode catalyst at 5 mA cm^−2^ by replenishing the Zn anode and electrolyte. Copyright 2017 Wiley-VCH Verlag GmbH & Co. KGaA, Weinheim [99].

**Figure 10 materials-15-05837-f010:**
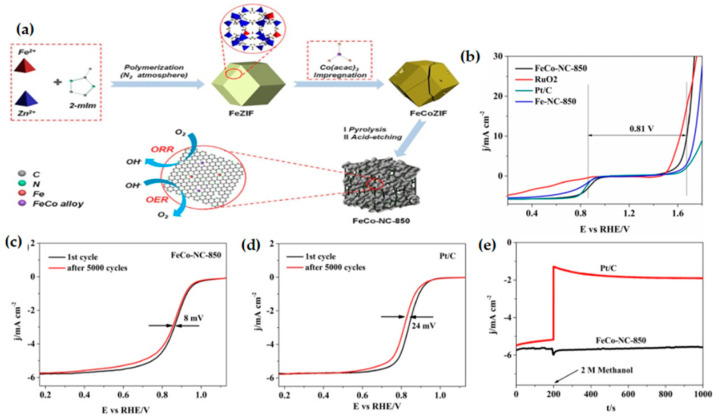
(**a**) Schematic illustration for the synthesis of FeCo-NC-850; (**b**) the overall LSV curves of FeCo-NC-850, Fe-NC-850, RuO_2_, and Pt/C at 1600 rpm in 0.1 M KOH. LSV curves of (**c**) FeCo-NC-850 and (**d**) Pt/C before and after 5000 cycles; (**e**) The i–t chronoamperometric response of FeCo-NC-850 and Pt/C with addition of 2 M methanol. All the catalysts were measured in 0.1 M KOH solution. Copyright 2019 Elsevier B.V. Allrightsreserve [107].

**Figure 11 materials-15-05837-f011:**
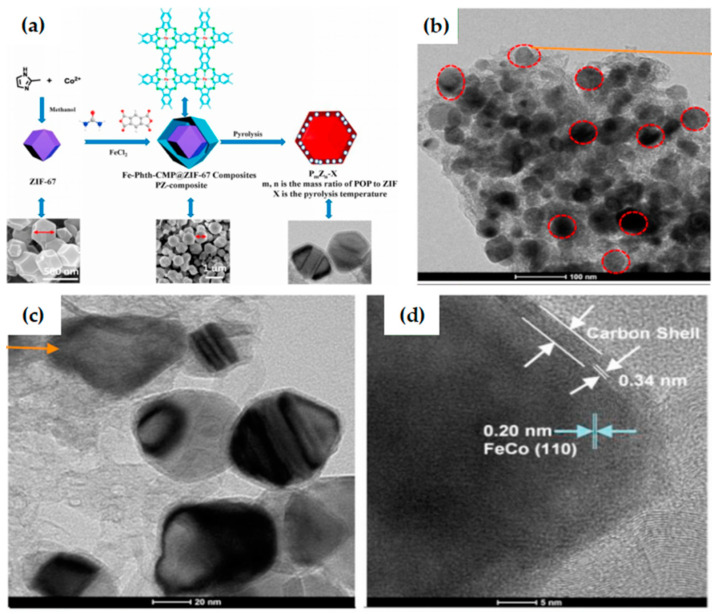
(**a**) Schematic of the process route for preparing P_m_Z_n_-X by solid phase synthesis method (m:n = 1:1, 3:2, and 2:3); TEM and HR-TEM of P_2_Z_3_-900: (**b**) TEM of P_2_Z_3_-900 at a scale bar of 100 nm; (**c**) TEM of P_2_Z_3_-900 at a scale bar of 20 nm; (**d**) HR-TEM of P_2_Z_3_-900 at a scale bar of 5 nm. Copyright 2020 Published by Elsevier Inc. [52].

**Figure 12 materials-15-05837-f012:**
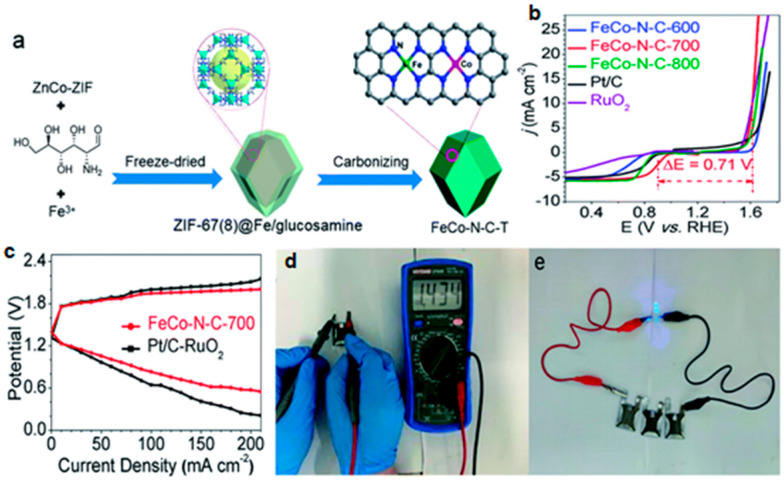
(**a**) Schematic illustration for the synthesis of FeCo-N-C-T; (**b**) ORR/OER bifunctional LSV curves of FeCo-N-C-T, Pt/C, and RuO_2_; (**c**) Discharge/charge polarization curves of rechargeable ZABs using FeCo-N-C-700 and Pt/C + RuO_2_ as the air electrode; (**d**) The open-circuit voltage of an all-solid-state ZAB; (**e**) The LED light powered by three all-solid-state ZABs. Copyright the Royal Society of Chemistry 2020 [109].

**Figure 13 materials-15-05837-f013:**
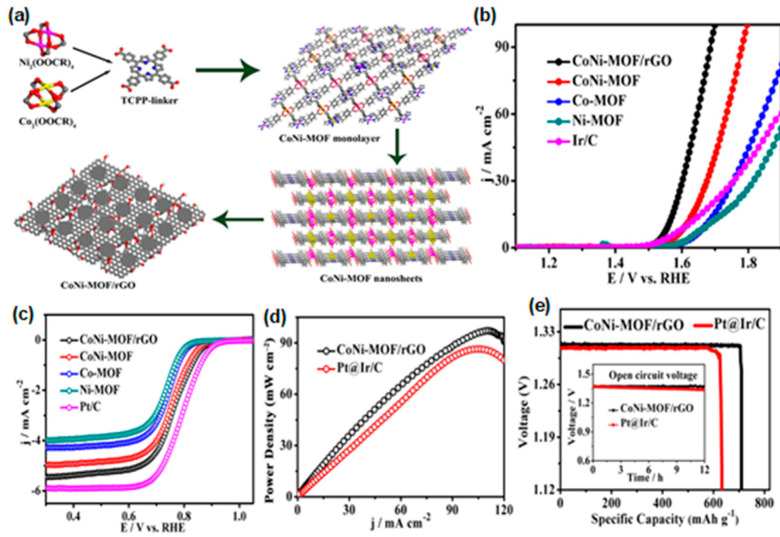
(**a**) Schematic illustration of the preparation of the CoNi-MOF/rGO catalyst; (**b**) OER LSV curves in N_2_-saturated 1.0 M KOH and (**c**) O_2_-saturated ORR polarization curves in 0.1 M KOH (CoNi-MOF/rGO, CoNi-MOF, Co-MOF, Ni-MOF and Ir/C or Pt/C electrocatalyst); (**d**) Corresponding power densities of CoNi-MOF/rGO- and Pt@Ir/C-based Zn–air batteries; (**e**) Corresponding specific capacity and open-circuit plot (inset) of CoNi-MOF/rGO- and Pt@Ir/C-based Zn–air batteries, normalized by the consumed weight of zinc. Copyright 2019 American Chemical Society [126].

**Figure 14 materials-15-05837-f014:**
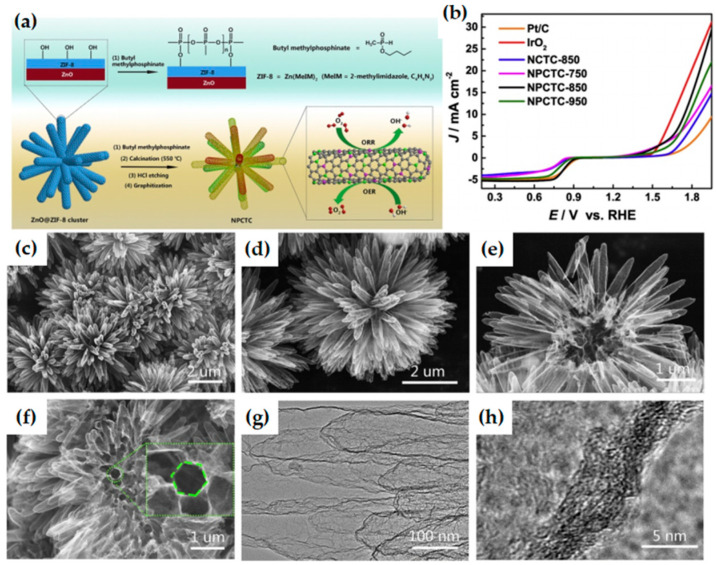
(**a**) Illustration of the preparation of NPCTC; (**b**) LSV curves of different catalysts for both ORR and OER at 1600 rpm in the whole ORR/OER region in O_2_-saturated 0.1 M KOH solution; (**c**–**f**) Typical SEM images of NPCTC-850; The inset in d shows the hexagon structural of single hollow carbon nanotube. (**g**,**h**) TEM images of NPCTC-850. Copyright 2018 Elsevier Ltd. All rights reserved [147].

**Figure 15 materials-15-05837-f015:**
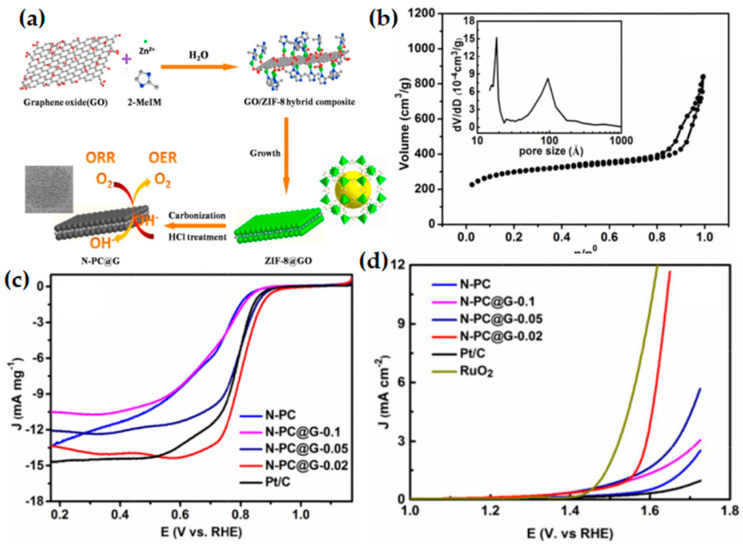
(**a**) Schematic illustration of a synthetic process of sandwich-like structured N-doped porous carbon@graphene (N-PC@G); (**b**) N_2_ sorption–desorption isotherm of the N-PC@G-0.02 sample; the inset of the corresponding pore-size distribution; (**c**) LSV curves of ZIF-8 derived carbon and commercial Pt/C catalysts in O_2_-saturated 0.1 M KOH solution at a scan rate of 10 mV s^−1^ and a rotating rate of 1600 rpm; (**d**) OER polarization currents of ZIF-8 derived carbon, RuO_2_ and Pt/C electrodes in 0.1 M KOH solution at a scan rate of 10 mV s^−1^. Copyright 2016 Elsevier Ltd. All rights reserved [54].

**Table 1 materials-15-05837-t001:** Summary of ORR/OER performance of MOF-based non-precious metal catalysts.

Catalyst	Electrolyte	ORR		OER	ΔE = E_j=10_ − E_1/2_	Ref.
		E_(onset)_ V vs. RHE	E_(1/2)_ V vs. RHE	E_(overpotential)_ at 10 mA cm^−2^		
Co-Nx/C	0.1 M KOH	\	0.877 V	300 mV	0.653 V	[59]
Co-MOF-800	0.1 M KOH	\	0.84 V	520 mV	0.84 V	[60]
NC@Co-NGC DSNC	0.1 M KOH	0.92 V	0.82 V	410 mV	0.82 V	[63]
Co_3_O_4_/HNCP-40	0.1 M KOH	\	0.845 V	350 mV	0.729 V	[65]
Co/CoO@NSC	0.1 M KOH	0.895 V	0.779 V	380 mV	0.775 V	[71]
NC-Co SA	0.1 M KOH	1.00 V	0.87 V	360 mV	0.72 V	[72]
FePPc@CB	0.1 M KOH	\	0.908 V	358 mV	0.68 V	[85]
Fe_3_C@NCNT/NPC	0.1 M KOH	1.0 V	0.9 V	270 mV	\	[88]
Fe-N-HPC-900	0.1 M KOH	1.004 V	0.886 V	520 mV	0.81 V	[51]
MnBDC@75% rGO	0.1 M KOH	1.09 V	0.94 V	610 mV	0.90 V	[95]

**Table 2 materials-15-05837-t002:** Summary of ORR/OER performance of MOF-based bimetallic catalysts.

Catalyst	Electrolyte	ORR		OER		BET Specific Surface Area	Ref
		E_(onset)_ V vs. RHE	E_(1/2)_ V vs. RHE	E_(overpotential)_ at 10 mA cm^−2^	Tafel Slope		
FeCo-NC-850	0.1 M KOH	0.997 V	0.864 V	445 mV	117 mV dec^−1^	553 m^2^ g^−1^	[107]
P_2_Z_3_-900	0.1 M KOH	0.950 V	0.807 V	370 mV	57 mV dec^−1^	153.22 m^2^ g^−1^	[52]
A-FeCoO@NCNs	0.1 M KOH	1.03 V	0.87 V	440 mV	80 mV dec^−1^	809.83 m^2^ g^−1^	[108]
FeCo-N-C-700	0.1 M KOH	0.013 V	0.896 V	370 mV	72 mV dec^−1^	332 m^2^ g^−1^	[109]
CoCx/FeCo@C	0.1 M KOH	1.018 V	0.965 V	390 mV	77.1 mV dec^−1^	\	[110]
B-FeNi-N/C-1000	0.1 M KOH	\	0.9 V	390 mV	283 mV dec^−1^	832.7	[120]
1.5FeNi@NCNT	0.1 M KOH	0.95 V	0.86 V	230 mV	55 mV dec^−1^	870.99 m^2^ g^−1^	[121]
FeNi-NCS-2	0.1 M KOH	\	0.867 V	395 mV	82.3 mV dec^−1^	454.77 m^2^ g^−1^	[122]
R-NCM	1 M KOH	0.90 V	\	319 mV	78.2 mV dec^−1^	\	[124]
CoN-MOF/rGO	1 M KOH	0.88 V	\	318 mV	48 mV dec^−1^	\	[126]
CoPNi-N/C	0.1 M KOH	0.93 V	0.84 V	310 mV	72 mV dec^−1^	446.8 m^2^ g^−1^	[127]

**Table 3 materials-15-05837-t003:** Summary of ORR/OER performance of MOF-based metal-free catalysts.

Catalyst	Electrolyte	ORR		OER	BET Specific Surface Area	ΔE = E_j=10_ − E_1/2_	Ref.
		E_(onset)_ V vs. RHE	E_(1/2)_ V vs. RHE	E_(j=10 mA cm_^−2^_)_ V vs. RHE			
BNPC-1100	0.1 M KOH	0.894 V	0.793 V	1.38 V	1348 m^2^ g^−1^	0.587 V	[146]
NPCTC-850	0.1 M KOH	0.92 V	0.83 V	1.74 V	912 m^2^ g^−1^	0.90 V	[147]
N-PC@G-0.02	0.1 M KOH	1.01 V	0.80 V	1.63 V	1094.03 m^2^ g^−1^	0.83 V	[54]

## Data Availability

All raw data in this study can be provided by the corresponding author on request.

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
