# Peer review of "Metal–Organic Frameworks (MOFs) Derived Materials Used in Zn–Air Battery"

_materials, 2022, doi:10.3390/ma15175837_

Round 1

Reviewer 1 Report

In this manuscript, recent advances in MOF-derived materials for Zinc battery applications were reviewed. The author focused on the MOF-derived single metal, bimetal, and nonmetallic catalytic systems involved in OER and ORR performances for zinc batteries. Overall, some problems should be revised before acceptance.

Comments:

1.     Some of the parts are just a compilation of data from the published articles.

2.     Detailed structure-activity relationship with respect to catalytic activity, catalyst poisoning, and catalyst regeneration should be discussed.

3.     Fabrication methods of MOFs derived materials for zinc-air battery applications should be discussed.

4.     What should the authors suggest to overcome the deactivation behaviors while designing the new catalyst?

5.     The cause of electrode poisoning should be discussed.

6.     The mechanism of MOFs-derived materials in zinc-air batteries should also be discussed.

Author Response

Response to Reviewer 1 Comments

In this manuscript, recent advances in MOF-derived materials for Zinc battery applications were reviewed. The author focused on the MOF-derived single metal, bimetal, and nonmetallic catalytic systems involved in OER and ORR performances for zinc batteries. Overall, some problems should be revised before acceptance.

Comments:

Point 1: Some of the parts are just a compilation of data from the published articles.

Response to the comment 1: Thank you for your valuable comments. We have added some discussion about the synthesis methods, structure-activity relationship, and mechanism as follows.

Line 304, page 8

  • Zhou et al. [64] prepared Co/CoO@NSC bifunctional electrocatalyst using Zn-MOF@Co-MOF as a template. The composite material with the interconnected porous structure exhibited excellent ORR/OER catalytic activity and stability.

Line 308, page 8

  • The hollow nanostructures were formed under high temperature. This hollow structure could improve the diffusion kinetics of the catalyst during ORR/OER. So, the catalyst shows excellent ORR/OER catalytic activity and high power density when it was used in the Zn-air battery.

Line 329, page 9

  • The pyrolysis of ZIF-67 is a facile and effective strategy to obtain porous Co3O4 nanoparticles [65.66]. Rrich oxygen vacancies and tetrahedral Co2+ can be found in these Co3O4 Because of the synergistic effect between CoOx species and nitrogen-doped carbon, this kind of catalyst exhibit excellent ORR/OER performance and good cycling performance in Zn-air battery.

Line 440, page 12

  • Because of the uneven distribution of Fe species in the precursor, heterogeneous structures and metal nanoparticles (NPs) were obtained by annealing Fe-doped MOF-derived catalysts[85.86]. The active metal NPs are covered by a thick carbon layer and hinder the catalytic process. Thus, it is necessary to find an efficient way to prepare catalysts with uniform Fe distribution. Besides, the carbon materials such as CNTs and graphene show high specific surface area and good electrical conductivity. So building MOFs/carbon materials composite is an effective way to improve catalytic performance [87,88].

Point 2: Detailed structure-activity relationship with respect to catalytic activity, catalyst poisoning, and catalyst regeneration should be discussed.

Response to the comment 2: Thank you for your valuable comments. We have added some descriptions about the structure and active substances.

Line 301, page 8

  • The simulation reveals that the intermediate (OOH*) has strong and favorable adsorption on the non-coordinating hollow-site C atoms relative to Co lattice, which may be accountable for the excellent bifunctional catalytic performance of the catalyst.

Line 385, page 10

  • The Co-N4 is always seemed as the active center of Co-based catalysts [72,73]. Because, the adsorption energy of O2 on CoN4 is low, the ORR could proceed with 2e- Thus, Xing and co-workers [74] proposed that Co2N5 active sites are active for the ORR. They considered that the Co-N path located at 2.12 Å in Co K-edge spectra corresponds to the structure of the bimetallic atom (Co2Nx). The Density functional theory (DFT) calculations revealed that the novel binuclear site exhibits considerably reduced thermodynamic energy barrier towards the ORR when compared to CoN4, thus contributing to enhanced intrinsic activity.

Line 560, page 14

  • Defining the accurate active site will facilitate optimal the framework of M-N-C catalysts. The unstable structure of M-N-C catalyst lead to insufficient stability. Here, we list some countermeasures:

Line 646, page 17

  • We changed the "N3Fe-CoN3" to "N4Fe-CoN4".

Line 659, page 17

  • DFT simulation show that Fe and Co in FeCo bimetallic catalysts affect the electronic structure, which has a synergistic effect on improving catalytic activity.

Point 3: Fabrication methods of MOFs derived materials for zinc-air battery applications should be discussed.

Response to the comment 3: Thank you for your valuable comments. We have reviewed the preparation methods of MOF-derived materials, and discussed the relationship between their unique structures and their catalytic performance.

Line 382, page 10

  • In conclusion, Co-based catalysts can be obtained by direct pyrolysis of Co-MOF, Co-doped ZIF materials, and ZIF-8@ZIF-67 core-shell materials. The catalytic performance of Co-based catalysts can also be improved by designing single-atom catalysts and carbon materials composites.

Line 470, page 12

  • In conclusion, MOF/CNT composites with efficient electrocatalytic performance were prepared. This provides a feasible way to prepare promising electrocatalysts.

Line 666, page 18

  • N-doped CoCx/FeCo@C/rGO catalyst shows excellent bifunctional catalytic performance. This depends on several things: (i) Atomically dispersed Fe and Co can provide more active sites. (ii) Rich N doping can modulate the electronic structure. (iii) The core-shell structure can increase the number of accessible active sites.

Point 4: What should the authors suggest to overcome the deactivation behaviors while designing the new catalyst?

Response to the comment 4: Thank you for your valuable comments. According to your suggestions, we have made following revisions.

Line 560, page 14

  • Defining the accurate active site will facilitate optimal the framework of M-N-C catalysts. The unstable structure of M-N-C catalyst lead to insufficient stability. Here, we list some countermeasures:

Line 570, page 14

  • The hollow structure can improve the stability of the catalyst.

Line 865, page 23

  • The following strategies are proposed to improve the activity: (i)Combining MOFs materials with carbon materials such as CNTs, graphene, , can effectively solve the problems of poor conductivity of catalysts and agglomeration of metal ions. (ii)Doping MOFs materials with transition metal ions can provide more active sites for catalysts. (iii)Designing catalysts with hollow structure, and core-shell structure can improve the mass transfer and expose more active sites. (iv)Designing single-atom catalysts or using template-assisted growth methods are also effective ways to increase active sites. (v)MOFs-derived metal-free nitrogen-carbon (NC) materials with large specific surface areas and high electrical conductivity can provide more charge transport channels. Doping MOFs-derived metal-free NC materials with heteroatoms (N, P, S, B, etc.) or combining them with graphene is a feasible method to improve the catalytic performance of MOFs-derived metal-free NC materials. Heteroatom doping can modify the local electronic structure of metal-free catalysts and effectively improve the catalytic activity. Graphene combining can improve the degree of graphitization of the catalyst, which is beneficial to electron transfer. In addition, the synergistic effect between NC derived from MOFs and N-doped graphene can provide more active centers.

Point 5: The cause of electrode poisoning should be discussed.

Response to the comment 5: Thank you for your valuable comments. We already pointed out the causes of electrode poisoning such as CO2 poisoning, zinc passiviation and dendrite formation in this review. In addition, we have added some suggestions for electrode poisoning in the "Summary and Outlook" section.

Line 895, page 24

In addition, optimization of cell configuration, electrolyte, and operating conditions is required to improve the performance of Zn-air battery such as power density, energy density, and stability. For example, we can choose appropriate additives to overcome zinc passivation and dendrite formation; Use filter based on chemical/physical absorption to reduce CO2 and prevent the deactivation of electrode materials caused by CO2 poisoning; Improve the utilization rate of electrolyte and prevent leakage.

Point 6: The mechanism of MOFs-derived materials in zinc-air batteries should also be discussed.

Response to the comment 6: Thank you for your valuable comments. According to your suggestions, we have added practical application and pointed out the mechanisms of MOF-derived materials in Zn-air battery.

Line 301, page 8

  • The simulation reveals that the intermediate (OOH*) has strong and favorable adsorption on the non-coordinating hollow-site C atoms relative to Co lattice, which may be accountable for the excellent bifunctional catalytic performance of the catalyst.

Line 387, page 10

  • Thus, Xing and co-workers [74] proposed that Co2N5 active sites are active for the ORR. They considered that the Co-N path located at 2.12 Å in Co K-edge spectra corresponds to the structure of the bimetallic atom (Co2Nx). The Density functional theory (DFT) calculations revealed that the novel binuclear site exhibits considerably reduced thermodynamic energy barrier towards the ORR when compared to CoN4, thus contributing to enhanced intrinsic activity.

Line 659, page 17

  • DFT simulation show that Fe and Co in FeCo bimetallic catalysts affect the electronic structure, which has a synergistic effect on improving catalytic activity.

Line 690, page 18

  • The Zn-air battery with FeNi-NCS-2 as the air cathode shows large open-circuit voltage, high power density, and excellent cycling stability.

Line 705, page 19

  • Moreover, the practical application in Zn-air battery is also better than that of Pt/C+IrO2/C.

Line 790, page 21

  • The Zn-air battery based on the NPCTC-850 air cathode exhibits a peak power density of 74 mW cm-2 and an energy density of 896 Wh kgZn-1, and has good charge-discharge cyclability.

Reviewer 2 Report

1. please specify in the caption of figure 4a, which material is described for the process.

2. "high activity and selectivity, excellent stability and conductivity, low cost and environmentally friendly." please change "friendly" which is not a noun.

 3. line 99 "The strategies,  mainly including different transition metal doping; single metal and bimetallic doping; 100 heteroatom doped MOFs derived metal-free electrocatalysts; and MOFs/graphene oxide 101 (GO) composites as bifunctional metal-free electrocatalysts." This sentence does not have a verb.

4.line 246. It will form a better structure if the authors add a transition paragraph between the mechanism and the following content about MOFs and introduce two kinds of MOFs discussed in sections 3 and 4.

5. I suggest that the authors draw a schematic that shows different categories of catalysts and/or the typical synthesis routes, which will help readers quickly understand the structure of the review.

Author Response

Response to Reviewer 2 Comments

Point 1: please specify in the caption of figure 4a, which material is described for the process.

Response to the comment 1: Thank you for your valuable comments. We have revised the caption of figure 4a as follows.

"Figure5. (a) Schematic diagram of the NC-Co SA fabrication process"

Point 2: "high activity and selectivity, excellent stability and conductivity, low cost and environmentally friendly." please change "friendly" which is not a noun.

Response to the comment 2: Thanks for your valuable comments. We are sorry for the mistake. We have revised this senetecnse as follows.

The main purpose of this review is to provide some feasible ideas for researchers to design environmental friendly bifunctional ORR/OER electrocatalysts with high activity and selectivity, excellent stability and conductivity.

Point 3: line 99 "The strategies, mainly including different transition metal doping; single metal and bimetallic doping; 100 heteroatom doped MOFs derived metal-free electrocatalysts; and MOFs/graphene oxide 101 (GO) composites as bifunctional metal-free electrocatalysts." This sentence does not have a verb.

Response to the comment 3: Thanks for your valuable comments. We are sorry for the mistake. We have modified this sentence as follows.

After that, some strategies to improve the activity and stability of MOF-derived ORR/OER electrocatalysts were summarized. The strategies are employed containing: (i) Transition metal catalysts derived from single and multiple transition metal-doped MOF, (ii) Metal-free catalysts derived from heteroatom-doped MOFs and MOFs/graphene oxide (GO) composites.

Point 4: line 246. It will form a better structure if the authors add a transition paragraph between the mechanism and the following content about MOFs and introduce two kinds of MOFs discussed in sections 3 and 4.

Response to the comment 4: Thanks for your valuable comments. We have added a paragraph before section 3.

Line 237, page 7

  • MOF-derived catalysts are one of the most promising bifunctional catalysts for ORR and OER. MOF-derived transition metal catalysts and MOF-derived non-metallic catalysts are illustrated in the following parts according the structural characteristics of MOF materials. We would like to build connections between performance of catalysts and their preparation strategies, and give some suggestions that will be helpful for the future development of MOF-derived ORR/OER catalysts. As shown in Figure 3, we present a classification diagram of MOF-derived for air cathode catalysts.

Point 5: I suggest that the authors draw a schematic that shows different categories of catalysts and/or the typical synthesis routes, which will help readers quickly understand the structure of the review.

Response to the comment 5: Thanks for your valuable comments. We have added a schematic diagram of the MOF material in page 7.

Figure 3. Classification diagram of MOF-derived catalysts for air cathode in Zn-air battery.
